# Evaluation capacity building in a rural Victorian community service organisation: A formative evaluation

Bianca E. Kavanagh[1,2]*, Vincent L. Versace[1], Hannah Beks[1], Carly Dennis[2], Marley J. Binder[1,2], Andrea L. Hernan[1], Kevin Mc Namara[1]

1 Deakin University, Deakin Rural Health, School of Medicine, Warrnambool, Victoria, Australia, 2 Brophy Family and Youth Services, Warrnambool, Victoria, Australia

* bianca.kavanagh@deakin.edu.au

## Abstract

Community service organisations are increasingly required to report on outcomes and evaluate program delivery. While commonplace in clinical health settings, such work is novel to the community sector and can be challenging to undertake given resourcing and evaluation capacity constraints. These constraints are exacerbated for rural community service organisations that face additional resource pressures and lack organisational systems to support such work. This formative evaluation reports on an evaluation capacity building program within a rural place-based community service organisation in southwest Victoria, Australia. As part of the program, monitoring, evaluation, and learning pilots were implemented to support selected teams (N = 4 teams, N = 12 individuals) to increase confidence in outcomes measurement and reporting. Implementation strategies included training, creation of measurement frameworks, implementation of data collection tools, academic support, and emergence of evaluation champions. Evaluation data included administrative records and a staff survey. The Reach Effectiveness Adoption Implementation Maintenance framework guided evaluation. Results indicated that participants reported high awareness, beliefs, attitudes, knowledge, and skills in relation to monitoring, evaluation, and learning at the post-training/refinement period timepoint. However, several barriers limited implementation: reduced workforce capacity, prioritisation of client-driven or other work, leave or staff changes, additional responsibilities, waxing and waning engagement, wider program teams being unclear on the process and value of evaluation, and organisational barriers. These barriers led to a divergence from the planned outcome measures. Enablers to this work included physical presence to facilitate informal discussion, flexibility and openness to change, and communication to support staff. Learnings include the need to strengthen organisational evaluation culture, scaling-back the number of outcomes/data collection sources, and greater involvement with the wider staff cohort. These learnings provide a valuable

**Data availability statement:** Data from the post-training/refinement survey can be found in the Supporting Information files. Administrative data cannot be shared publicly as they contain potentially identifying information. These data are available from Brophy's Quality and Compliance team at quality@brophy.org.au

**Funding:** BEK is funded by Brophy Family and Youth Services and the RHMT. VLV, HB, and MJB are also funded by the RHMT. The funders did not play a role in the study design, data collection and analysis, decision to publish, or preparation of the manuscript.

**Competing interests:** The authors have declared that no competing interests exist.

foundation for other place-based community service organisations to implement evaluation capacity building strategies.

## Introduction

### Background and rationale

Community service organisations (CSOs) are central to the prevention, management, and treatment of a range of social and health issues. CSOs provide support to maintain and foster the functioning of individuals, groups, and families, to increase their potential and enhance community wellbeing [1]. Within Australia, CSOs are not-for-profit and non-government organisations which are contracted by the government to provide social services through the delivery of material support and assistance to the population [2]. They fit within the broader social service, social care, or community care context globally. Although not nationally approved, some consensus exists of what a CSO's purpose is, and they tend to offer services that include disability support programs, aged care, child protection, and youth justice programs [1]. Often, there is an overlap with health and housing/homelessness services [1].

CSOs are largely funded by a competitive tender process through State and Territory governments [3]. Lesser amounts of funding may also be derived from the federal government, as well as via commercial and philanthropic sources [3]. Funding bodies tend to govern the programs a CSO delivers, while the organisation itself is governed by a Board of Directors. Particular programs also have mandatory accreditation requirements against standards (e.g., the Social Service Standards), while other accreditation requirements are voluntary and can be achieved to demonstrate a commitment to providing safe and quality care. The decisions and implementation concerning the delivery of services is placed at the community level, with local government or civil society [4]. Although limited recent data are available, it is estimated that between 2014–2015 the social services sector contributed ~$11 billion to the Australian economy, employing >143,000 full time equivalent staff and >590,000 volunteers [5]. These figures have likely increased since the implementation of the National Disability Insurance Scheme (NDIS] in 2016 and suggest that these services represent considerable importance to Australian society.

Increasingly, CSOs—and not-for-profits organisations more broadly—are required by funding bodies to report on outcomes (i.e., the effect on a participant during or after their involvement in an activity delivered by a CSO) and undertake evaluations. This is a change from traditional requirements, where reporting requirements tended to include only the reporting of outputs and the provision of audited financials [6–8]. This change in reporting perspective brings with it a need to focus on more than levels of activity; organisations must plan to implement activities in a way that optimises outcomes within their service delivery context. It has also seen the need to create and share organisational knowledge, however literature on organisational learning and knowledge is an underrepresented area in the literature for not-for-profit organisations [9]. Organisational learning refers to the collective learning process that

is regulated in an organisation to improve organisational performance and goals, affecting an organisation's processes, structures, routines, and future learnings [10]. Crossan et al's model of organisational learning suggests it is dynamic in nature; integrates individual, group, and organisational levels of analysis; incorporates exploration and exploitation tensions; and is open to various types of changes grounded in experience [11]. The model suggests that organisational learning occurs via bidirectional intuiting, interpreting, innovation, and influence processes that operate across the individual, group, and organisational levels. Barriers to organisational learning can therefore be targeted towards these processes [10].

Monitoring, evaluation, and learning (MEL) can support CSOs to understand the context of their work, and inform service adaptations, organisational strategies, and initiatives for scaling up [12], and may also promote organisational learning. To support this, organisations need to have responsive processes, accessible resources, and personnel who are committed to the value of the work [13]. MEL is utilised across many organisational contexts, including in policy documents, non-governmental reports, and research. In a diverse range of settings including CSOs, agriculture and international aid, MEL supports increased accountability, adaptability, change optimisation and learning where it is desirable to understand the impact of innovation and knowledge translation activities designed to support strategic organisational and system changes [14–18]. Some MEL frameworks utilised by organisations are made publicly available to support transparency (e.g., [11,12]. Oftentimes, the term 'monitoring' is used interchangeably with 'measurement', however, monitoring may be perceived as being more active in its approach to using and learning from data insights, and thus is the approach discussed in this paper.

Fostering an organisation's capacity to undertake MEL can be achieved via evaluation capacity building—defined as the intentional effort to continuously generate and sustain overall organisational processes and equipping staff with the skills that enable quality evaluation which becomes part of routine practice [19,20]. Evaluation as a field of research is receiving increased recognition for its primary purpose of collating information with the ultimate use being in decision making [21]. Evaluation capacity building can support organisations to become evidence-informed, respond to increased requirements to conduct evaluations, and facilitate decision-making [22]. A multifaceted approach, including through training, technical assistance, and mentoring can support this [22].

An emerging area which may be beneficial to MEL and evaluation capacity building is the implementation of practice research or embedded research. Practice research is entrenched in social services [23], whereby social work researchers and practitioners work together to identify, understand, and improve service delivery and organisational structures and processes. McBeath et al [24] recommend that collaborative, trust-based relationships should be developed and sustained to achieve practice research. Further, sustained self-reflection including deliberation of the tension between the type of research generated (e.g., empirical peer-reviewed research vs grey literature, agency-based statistics) as well as the expectations of organisational staff, service users, and other stakeholders should occur to maintain varied relational commitments [24]. Embedded research also aims to build the capacity of an organisation to conduct research via the embedding of a researcher into a host organisation. The embedded researcher is generally required to developed relationships with staff at the host organisation and be seen as a member of the team and to generate knowledge that meets the needs of the host organisation [25]. Embedded research, however, has traditionally been conducted within healthcare settings, and is an emerging area of research in other settings, including within CSOs.

Evaluation capacity building, embedded research, and practice research have not been a strong focus of the resource-constrained CSO sector in the past [4], and this sector may benefit from lessons learned in the broader social work sector. Further, basic indicators to monitor evaluation capacity building have not been well-developed [4] and there is no agreed measurement and evaluation framework for not-for-profit service outcomes in Australia [26]. Barriers (e.g., resource, infrastructure, workforce capacity, and knowledge limitations) have also led to challenges for staff to incorporate MEL as part of their routine practice [27]. This culminates in instances where funding contracts mandate specified output and outcome reporting, yet CSOs do not have sufficient data collection software to record and collate the required

information [3]. To support this, CSOs must have research-minded practitioners who are interested in and utilise evidence to inform practice. Research practitioners or embedded researchers may support practitioners to become research-minded, through their assistance and leadership in developing processes and building capacity to collaboratively evaluate processes and create practice-based solutions [28,29].

Due to the paucity of evaluation evidence produced by CSOs at the local level, organisations may not have sufficient evidence-informed data to enhance processes and outcomes, meet the requirements of funding agreements, and to learn, improve, and meet the needs of clients [30]. Further, there is limited evidence detailing the effective implementation and sustainability of continuous quality improvement [13], outcome measurement, and MEL [31]. This lack of evidence is even greater in rural areas which do not have the same resources, capacity, or capability as metropolitan areas to produce local evidence [32]. These constraints may be exacerbated for rural CSOs that face additional resource pressures and lack infrastructure to support such work. Additionally, rural areas in Australia are almost invariably of lower socio-economic position than metropolitan areas [33]. For these reasons, it is important to share the experiences of developing and implementing a program of local-level evaluation capacity building within a place-based CSO. The aim of this work was to conduct a formative evaluation of MEL implementation in order to offer practical guidance for early-stage implementation by other CSOs.

## Study context

Brophy Family and Youth Services (Brophy) is the primary place-based provider of family and youth services in southwest Victoria, Australia. Its services are delivered from offices in Warrnambool, Portland, and Hamilton, as well as outreach services across the region. The southwest region is encompassed by a large rural town (Modified Monash Model [MM] 3), and medium and small rural towns (MM 4 and MM 5) [34], and comprises large agricultural areas.
Brophy's Executive Leadership Team identified a need to generate in-house local-level research and evaluation to enable the inclusion of MEL as an eventual part of usual business; support competitive tender processes; provide timely data on program outcomes; and to establish a culture of evidence-informed practice more broadly (see Kavanagh et al for further information [35,36]). This need was identified following the recognition that many tender processes were beginning to include evaluation as a requirement. To achieve this, a program of evaluation capacity building work was designed by members of Brophy's Executive Leadership Team, Quality and Strategic Projects Manager (CD), and Research and Evaluation Lead. Additionally, members of Deakin Rural Health (a University Department of Rural Health) also had input into the monitoring and evaluation of the capacity building program. The program considered the constraints of the place-based rural context, including that staff have low exposure to research and evaluation opportunities, including training and career development, and limited infrastructure to support research and evaluation. It included supporting the implementation of MEL, supporting bespoke outcomes evaluations as required by funders, identifying evaluation champions, and including evaluation activities in relevant organisational documentation. MEL was selected as a primary focus for this capacity building program, and an initial pilot approach was chosen due to limited resourcing, little evidence to guide MEL implementation in CSOs, and to identify issues with the implementation process, as well as the feasibility and practicality of the work. An experiential learning approach was utilised, which included contextually rich concrete experiences, contextually-specific critical reflection and abstract conceptualisation, and pragmatic active experimentation [37].

All 38 client/stakeholder programs at Brophy were considered for participation in the MEL pilots. Between January and March 2023, Brophy's Executive Leadership Team selected four programs to pilot MEL based on the perceived priority need for MEL (i.e., programs that were either newly delivered, had a need to better understand the outcomes their clients were achieving, or had funding requirements that necessitated data to be captured and reported). These programs included services for individual and family support, alcohol and other drugs, foster care, and youth homelessness and employment. Pilot team members (i.e., staff representatives who are taking part in the MEL pilots who belong to a wider

team of staff who are employed within a specific program of work) were selected based on their role and interest in MEL. Each team included an Executive Manager, Manager, and a Team/Practice lead.

During 2023, pilot teams undertook training in MEL. Training and workshops were delivered online via an external consulting firm (due to the absence of research and evaluation team in place at Brophy at the time) and on (1) understanding measurement; (2 and 3) creating a theory of change; (4) undertaking effective measurement; and (5) developing a measurement framework. Workshops (approximately two hours each) were delivered four times across a fifteen-week period, and teams had several weeks in between each workshop to complete accompanying reading and training materials. Based on learnings from the training and workshops, paired with experience garnered through practice, pilot teams developed overarching key evaluation questions, a theory of change relevant to each of their programs, and drafted outcome measures during the workshops (i.e., contextually rich concrete experience). In addition, a MEL Lead Group—comprised of each MEL pilot team and members of Brophy's Executive Leadership Team—was established and pilot teams met regularly to discuss MEL pilot implementation (i.e., critical reflection and abstraction conceptualisation).

At the beginning of 2024, pilot teams worked with Brophy's newly established Research and Evaluation Lead (BEK), a Research Fellow with expertise in mental health research and evaluation, to refine outcome measures and develop measurement frameworks and protocols. Pilot teams met regularly during this refinement processes (i.e., between three and six meetings, depending on the needs of the pilot team) and corresponded regularly via email to troubleshoot issues (i.e., critical reflection and abstraction conceptualisation). Implementation strategies support the MEL pilots included the creation of measurement frameworks, implementation of data collection tools, and academic support, and emergence of evaluation champions. Importantly, as little was known about the feasibility and practicality of MEL in practice, the role of evaluation champion was informal, and its function was observed (rather than expected) in champions' motivation, responsiveness, and advocacy. An overarching theory of change was also developed during this time to support this work (see Fig 1).

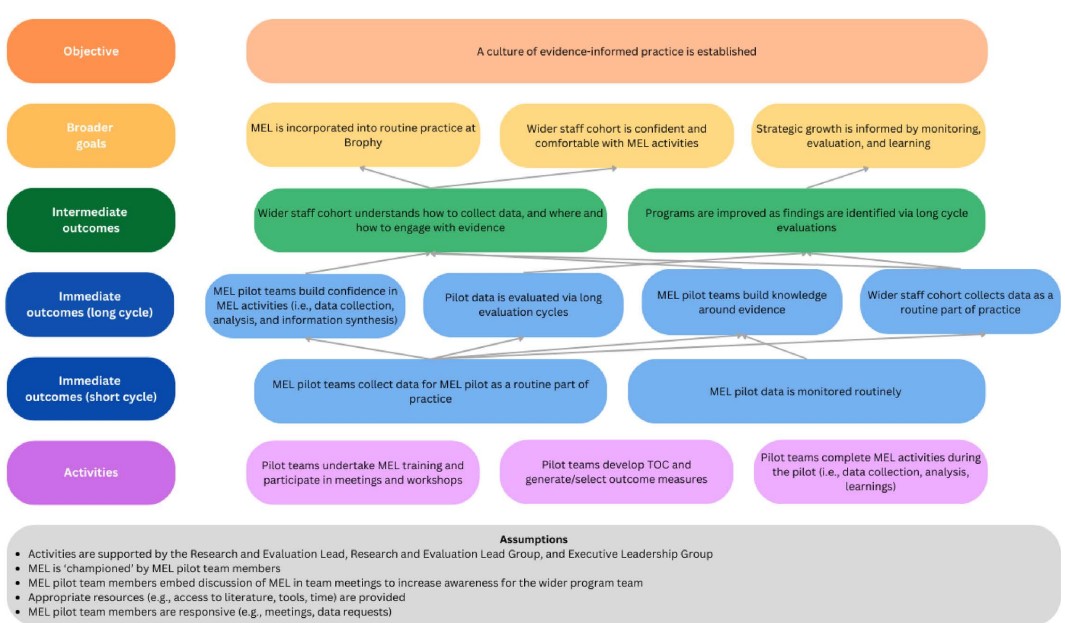

**Fig 1. Theory of change for MEL pilots.**

Data collection for the MEL pilot began on the 29th of April 2024. A short and long cycle evaluation method was selected, with short cycles spanning quarterly intervals, and the long cycle spanning the two-year period. The purpose of the short evaluation cycle (SEC) is to monitor data, rectify any identified issues with implementation, increase staff confidence with collecting and reporting on data, and to apply these learnings by making modifications to the implementation of the following SEC (i.e., pragmatic active experimentation). Although the conclusion of each SEC provided an opportunity to make changes to implementation, teams were also encouraged to discuss and implement desired changes throughout SECs—this was a crucial way to allow learnings, problem-solving, and adaptions to occur.

Formal evaluation against prespecified key evaluation questions will occur at the end of the two-year pilot period, and an interim evaluation is planned in 2025. A formative evaluation, which forms the basis of this paper, has also taken place to understand the mechanisms and context affecting MEL pilot implementation, so that learnings can be made and applied to the broader evaluation capacity building program. The formative evaluation considers the first two short SECs (i.e., first six months) of MEL pilot implementation. Further details on the MEL pilot process are presented in Fig 2.

## Materials and methods

Formative evaluations are undertaken to identify potential and actual influences on the progress and effectiveness of implementation efforts, study the complexity of implementation projects, and propose methods to answer questions concerning context, adaptations, and change responses [38]. The purpose of the current formative evaluation is to understand the mechanisms and context affecting MEL pilot implementation, so that learnings can be made and applied to the broader evaluation capacity building program. The RE-AIM framework is a widely used framework for assessing the impact of programs and was applied to the current evaluation. It includes key dimensions of reach, effectiveness, adoption, implementation, and maintenance [39,40]—see S1 Table. The key dimensions of RE-AIM function are measured

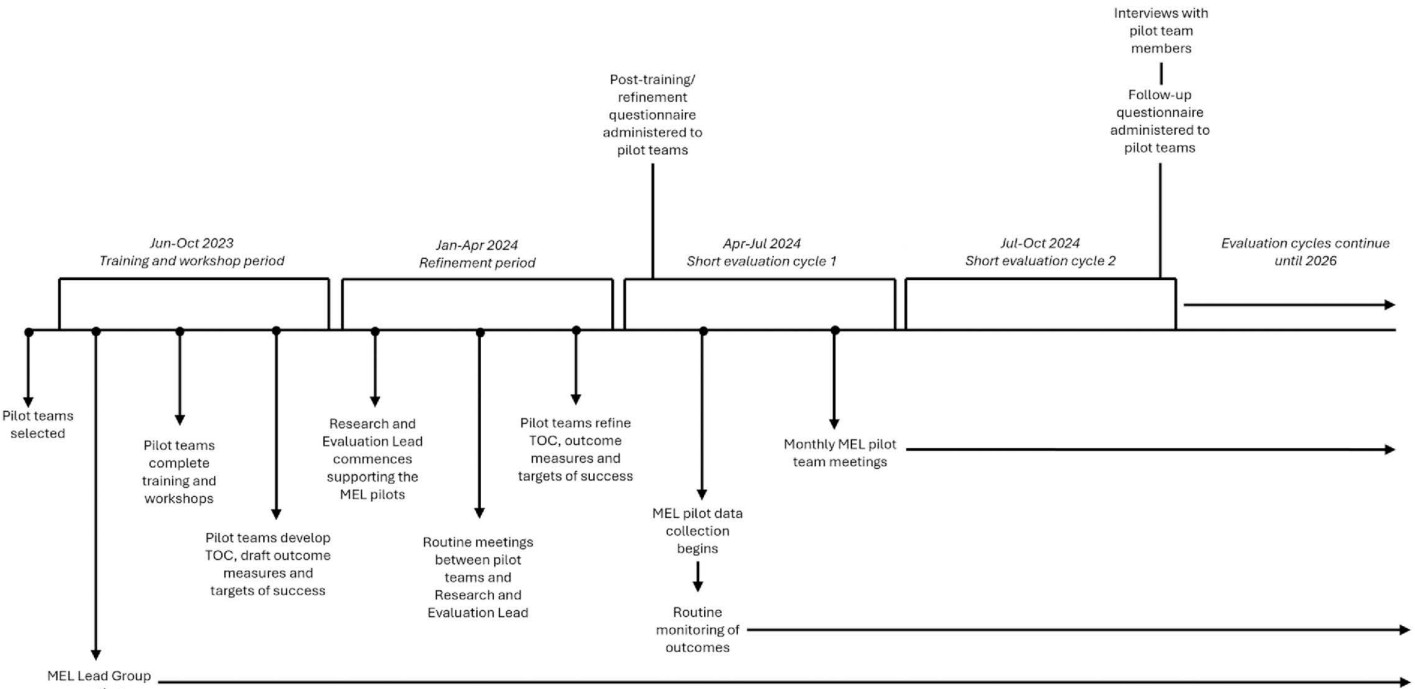

**Fig 2. Planned timeline of MEL pilot and formative evaluation.**

at the individual (i.e., reach, effectiveness, and maintenance) and multiple ecological levels, including the staff, setting, system, policy/other levels (i.e., adoption, implementation, and maintenance) [40]. As the current evaluation considers implementation over the first six months, maintenance is still in an early phase and results represent initial efforts towards maintenance. RE-AIM key dimensions and related outcome measures and method(s) of assessment can be found in Table 1.

The purpose of this article is to report the formative evaluation findings relating to the reach, effectiveness, adoption, implementation, and maintenance of the MEL pilots—representing an element of a broader evaluation capacity building program. Recruitment for the post-training/refinement questionnaire occurred between 09/05/2024 and 08/06/2024. Recruitment for the follow-up questionnaire and interviews took place between 31/10/2024 and 30/11/2024. Prior to the collection of any evaluation data, all staff involved as participants in the questionnaires provided voluntary online informed consent via ticking a checkbox. Written informed consent was obtained for interview participants. This study has received Deakin University ethics approval (reference number: HEAG-H 47_2024).

## Data sources

**Questionnaires.** A self-report questionnaire was administered to participants (i.e., MEL pilot members) at the beginning of the first SEC, following the training and refinement period. This questionnaire was adapted from a structured questionnaire on evidence-informed practice [41], by replacing reference to 'evidence-informed practice' with 'MEL'. Additionally, the tool was adapted by omitting two of the original follow-up questions concerning 'adding new' and 'confirming' clinical decision making. The adapted tool included demographics (i.e., length in role, level of position, employment status, and highest education qualification) and questions concerning MEL awareness, beliefs, attitudes, knowledge, skills, and implementation (see S1 File). Responses were rated on a five-point Likert scale from *one (not at all)* to five *(yes, to a large degree)*. The question concerning implementation was rated on a three-point Likert scale

**Table 1. Key dimensions of RE-AIM, and formative evaluation outcome measures and methods of assessment.**

| Key dimension | Outcome measure(s) | Method(s) of assessment |
|---|---|---|
| Reach | Number of programs at Brophy that were considered for MEL pilot implementation | Administrative data documenting consideration of MEL pilot teams at Executive Leadership Group meetings |
| | Characteristics of the participating population (i.e., MEL pilot teams) compared to the non-participating population (i.e., programs that did not participate in the MEL pilots) | Administrative data on types of programs and their resources for MEL |
| Effectiveness | Pilot teams self-reported data on the awareness, beliefs, attitudes, knowledge, and skills concerning MEL | Post-training/refinement and follow-up survey results of MEL pilot teams |
| Adoption | Pilot teams that completed the training and attended relevant meetings during the refinement period | Administrative data on the number of pilot teams that attended MEL training and meetings |
| | Number and reasons for pilot team members seeing a need to implement MEL | Qualitative interviews with MEL pilot team members |
| Implementation | Number of implementation activities that were administered according to the relevant protocol and measurement framework | Administrative data on implementation activities, including meetings during refinement and short evaluation cycle periods, and adherence to planned outcome measures and data collection methods |
| | Pilot teams self-reported data on the experience and skill changes over time concerning the implementation activities | Post-training/refinement and follow-up surveys of MEL pilot teams; qualitative interviews with MEL pilot team members |
| | Number and type of barriers to MEL implementation | Follow-up survey of MEL pilot teams; administrative data on MEL implementation activities; qualitative interviews with MEL pilot team members |
| Maintenance | Number and type of changes to policies, practice documents, and organisational documents to support MEL | Administrative data; follow-up survey results of MEL pilot teams; qualitative interviews with MEL pilot team members |

from *one (no)* to *three (yes)*. The same questionnaire—with an additional two qualitative questions concerning practice decision-making and barriers to MEL implementation—was administered at the conclusion of the second SEC.

**Interviews.** Qualitative semi-structured interviews were conducted in-person with participants at the conclusion of the second SEC. Questions were guided by the Consolidated Framework for Implementation Research (CFIR)—a widely used framework to assess existing/potential barriers and enablers to successful implementation (i.e., innovation, outer setting [context], inner setting [context], individuals, and implementation processes) [42,43]. Questions covered participants' background in relation to the MEL pilot, pre-implementation experiences, implementation experiences, and impact. Qualitative data from interviews were audio-recorded and transcribed using Microsoft Word. The interview guide can be found in S2 File. Interviews were conducted with Brophy's Research and Evaluation Lead (first author).

**Administrative records.** Administrative records were also utilised to gauge the implementation of the MEL pilots. These records included the protocol and measurement frameworks that were developed for each pilot team, as well as Microsoft Excel, Forms, and List documents which were set-up for MEL pilot data collection, analysis, and presentation. In addition, minutes from meetings, emails, phone/video calls, and a meeting progress tracker were used to assess the implementation of this work.

### Data analysis

This evaluation used a mixed methods study design. Data for the analysis were derived from the post-training/refinement survey and administrative data collected throughout the pilot period to date. Nil qualitative interview data was included in the analysis. Quantitative data from the post-training/refinement survey were collected in Qualtrics XM and analysed using Microsoft Excel v2410 and reported as *n* (%) or Mean (SD) as relevant. Microsoft Excel, Forms, and List documents were assessed against each team's protocol and measurement framework as having been administered according to planned timeline, outcome measures, and reporting requirements, and these items were discussed at MEL pilot meetings. High level themes that related to the implementation of the MEL pilots were extracted from meeting minutes. Adherence was measured as outcome measures being administered according to the protocol and measurement framework (yes/no). Deviations occurred when an outcome measure had not been administered at all. The number of meetings each pilot team had within the SECs was cross-tabulated and reported. The data analysis originally planned to include triangulation of interview data, quantitative data, and administrative records but this analysis was unable to be performed due to lack of interview data.

### Results

#### Reach

In total, N = 11 team members were originally selected across N = 4 MEL pilots. During the data collection period, an additional team member was added to one pilot team which was originally comprised of two members, bringing the total to N = 12 pilot team members. Each pilot team was a subset of part of a wider program team. These wider program teams included staff who were not directly involved in the MEL training and workshop period or refinement period but were indirectly involved in data collection during the short evaluation cycles (see Fig 3). Pilot teams were situated across two offices (Warrnambool and Portland) and spanned service delivery for children, adults, and families. Each program differed in their aim, funding, and structure; however, overall, pilot teams did not considerably differ from the teams that did not participate, with the exception that some non-participating teams either already had processes in place for collecting, monitoring, and analysing data, or were not in a position to undertake MEL at the time of implementation. Each participating team was engaged in all aspects of the MEL pilot (i.e., training and workshop period, refinement period, SEC 1, SEC 2).

Results from the voluntary post-training/refinement questionnaire (n = 6; 50% response rate) revealed that one participant was a Practitioner, two participants were Team/Practice Leads, two participants were Managers, and one was an Executive Manager. All participants worked full time. Four participants had a diploma/trade certificate, and two participants had postgraduate degrees (see S3 File).

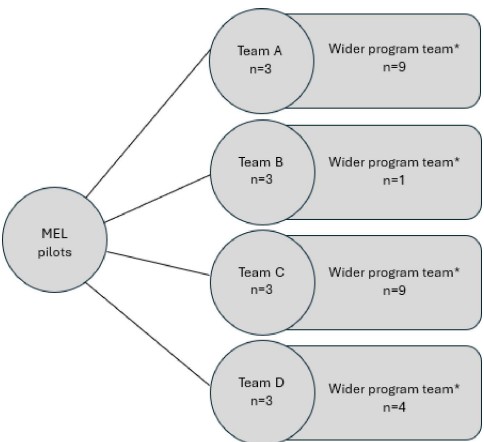

**Fig 3. Composition of MEL pilot teams in relation to their wider program teams.**

## Effectiveness

The effectiveness of the MEL pilots was considered in relation to participants' awareness, beliefs, attitudes, knowledge, skills, and implementation of MEL. At the post-training/refinement timepoint, the Mean for each concept was above four (range one to five), suggesting that participants had a high self-reported awareness, beliefs, attitudes, knowledge, and skills in relation to MEL, and that most participants had searched for relevant literature in the prior six months (implementation). Results from the post-training/refinement survey are displayed in Table 2. Nil responses were received for the follow-up survey, and only one participant agreed to be interviewed concerning their experience of the MEL pilot to date and as such these data were not able to be included.

## Adoption

All four teams (100%) completed the MEL training and adopted the MEL pilots during the training and workshop period. Due to the differing nature, structure, and resources of each pilot team, coupled with an aim to build research and evaluation capacity at Brophy (see Kavanagh et al for further information 35,36), a bespoke approach to routine meetings between each pilot team and the Research and Evaluation Lead was undertaken during the refinement period. These meetings facilitated developing and refining the MEL pilot protocols, measurement frameworks, outcomes, and data collection tools. Discussion via email and Microsoft Teams also facilitated this process. This approach allowed MEL pilot teams to meet as often or as little as needed during this time, with a requirement that some input from each team was required to progress with data collection. Meeting frequency also depended on the complexity of the data collection tools

**Table 2. Post-training/refinement survey results concerning MEL concepts.**
**Data reported as Mean (SD) or n (%).**

| Concept | Post-training/refinement (n = 6) |
|---|---|
| Awareness | 4.3 (0.82) |
| Beliefs | 4.8 (0.41) |
| Attitudes | 4.8 (0.41) |
| Knowledge | 4.0 (0.89) |
| Skills | 4.3 (0.82) |
| Implementation (yes) | 4 (66.7%) |

and reporting process. Analysis of administrative data revealed that during the refinement period, meeting frequency ranged between two and five meetings (see Table 3).

## Implementation

During SECs one and two, a monthly meeting with each pilot team was planned to support the collection, extraction, and monitoring of data. Monthly meetings were designed to support the monitoring process and to troubleshoot any issues with data collection and reporting. Administrative records, including the frequency of missed meetings, meeting minutes, and email communications between the Research and Evaluation Lead and pilot teams revealed that several barriers reduced the frequency of meetings, which in turn impacted the planned implementation of the MEL pilots. These barriers included MEL pilot members having limited capacity to dedicate to MEL within their role, taking planned or unexpected leave, acting in higher positions with additional responsibilities, and needing to prioritise client-driven or other work. These barriers led to a waxing and waning of engagement with the MEL pilots in the first six months of this work, which had a direct effect on data collection and reporting, leading to a divergence from the planned protocol and measurement frameworks. In addition, it was identified that including some but not all program staff in the initial training and refinement period may have resulted in limited clarity regarding the process and value of MEL, further affecting data collection.

Organisational barriers also indirectly affected implementation. These included unexpected changes to funding resulting in significant operational change during the evaluation period; suboptimal routine documentation of service delivery; shortcomings in governance and reporting mechanisms for evaluation activities; and an underdeveloped evaluation culture. Further, the change from reporting on funding targets to program outcomes was a more novel organisational change than expected, and this led to a heavier than anticipated reliance on the Research and Evaluation Lead for data monitoring, extraction, analysis, and reporting. Overall, these barriers resulted in a lower than planned number of outcome measures being used for data collection. Across the pilot teams, adherence to the planned outcome measures ranged between 50.0% to 88.9% (see Table 4).

Despite the above issues, several enablers to implementation were also identified. These enablers included physical presence to facilitate discussion between the Research and Evaluation Lead and MEL pilot team members, being flexible and open to changes, and written and verbal communication to aid staff understanding in the process and requirements of the MEL pilots.

## Maintenance

Preliminary efforts towards maintenance—including the sustainability and efficiency of MEL—have occurred despite the initial stages of this work. Discussions throughout the first SEC and reflections upon its conclusion revealed the need to scale back the number of outcome measures being used for data collection to reduce the workload for three teams. For instance, Pilot Team D opted to remove three of their client-based surveys at the end of SEC 1. These surveys measured outcomes that were either able to be captured by other data sources (i.e., client record management system) without adding additional burden to staff or clients or were deemed to be less important than other outcomes on their theory of change.

**Table 3. Number of meetings during the refinement period and SECs one and two.**

| Team | Refinement period | SECs one and two (combined) |
|---|---|---|
| Pilot team A | 3 | 2 |
| Pilot team B | 2 | 3 |
| Pilot team C | 5 | 5 |
| Pilot team D | 3 | 4 |

**Table 4. Divergence of data collection and reporting from protocol and measurement framework.**

| Pilot Team | Number of outcomes | Number of outcome measures | Number of scales/ domains measured | Adherence to outcome measures | Adherence to planned reporting |
|---|---|---|---|---|---|
| A | 3 | 9 | 14 | 89% | Complete for SEC 1 and SEC 2 |
| B | 5 | 19 | 28 | 74% | Complete for SEC 1 and SEC 2 |
| C | 2 | 2 | 6 | 50% | Complete for SEC 1 and SEC 2 |
| D | 5 | 6 | 9 | 50% | Complete for SEC 1 and SEC 2 |

Note: SEC = short evaluation cycle

To help support evaluation capacity building broadly, as well as the embedding of MEL at the organisation level, reference to participation in evaluation activities has now been included in all position descriptions. In addition, work has begun to include evaluation activities within organisational supervision documents and some team-specific meeting documents.

## Discussion

This formative evaluation reported on the implementation of MEL pilots as part of a broader program of evaluation capacity building at a rural CSO in Victoria, Australia. Results revealed that despite participants having high levels of self-reported awareness, beliefs, attitudes, knowledge, and skills in relation to MEL following the training and refinement period, several barriers prevented the MEL pilot implementation to its full potential, while some facilitators supported the work. Barriers included having limited capacity within roles to dedicate to MEL, prioritisation of client-driven or other work, leave, additional responsibilities, waxing and waning engagement, staffing changes, wider program team members being unclear on the process and value of MEL, and unforeseen organisational issues that presented during the pilot. Facilitators to this work included physical presence to enable informal discussion, flexibility and openness to change, and communication to support staff. It is important to note, however, that current results are largely based on administrative data due to the absence of follow-up survey and interview data, and as such, may not be generalisable to the wider CSO or similar context.

At the conclusion of this formative evaluation, the short-term outcomes of the current MEL pilot as per the theory of change were still emerging. The embedding of data collection within practice was still developing at this study's completion and may take several more SECs before it is considered routine practice. Formal data monitoring did not occur as routinely as originally planned (i.e., monthly), however, each team monitored their data at the end of each SEC. In addition, several data collection issues (e.g., data entry mistakes) or ideas were identified by MEL pilot teams on many occasions, and as such, informal data monitoring has occurred regularly. It may be that an informal approach (i.e., when pilot team members have spare time) paired with quarterly meetings to monitor data at the end of each SEC are better suited to this context. As the long evaluation cycle is planned for a two-year period, it is not possible to comment on whether the intended immediate outcomes relating to this cycle have been achieved, however, researcher observation suggested that confidence among MEL pilot team members is building.

Some assumptions of the theory of change were not met in their entirety. Time pressure was a challenge as teams did not have protected time to dedicate to MEL implementation. Limited time, paired with the aforementioned barriers, had flow-on effects to responsiveness. Additionally, MEL being embedded within wider team meetings (e.g., as an agenda item) to increase awareness for staff not directly involved in the pilot was not established at the commencement of the first SEC. However, this has since been addressed and included as a discussion point for pilot teams as the pilot has progressed.

Additionally, it was an oversight to not include input from the organisation's Business System Coordinator in the planning stage (i.e., training and workshop period) of this work; this should have been included as an important resource

within the theory of change. The Business System Coordinator's support was critical in mitigating challenges with some aspects of data extraction and reporting during the refinement and data collection periods. This was especially the case when reports were derived from complex systems that were new to the organisation and MEL pilot team members had less familiarity and confidence in their potential use. As such, this support has been recognised as being essential to the efficiency and sustainability of the MEL pilots going forward.

### Links to the broader literature

Although evaluation capacity building is a relatively new field (while research capacity building is more established), our results are largely in line with the limited literature available documenting the challenges and enablers of evaluation practice within community organisations [22,44–47]. Of the few studies detailing the impact of evaluation capacity building in community organisations, Satterlund and colleagues [44] suggested that the voluntary nature of their evaluation capacity building program limited active buy-in and engagement with program personnel. This barrier may be common as staff tend to be focussed on interventions rather than evaluations [44]. Although the current MEL pilots weren't voluntary in the same way, engagement was up to individuals/teams and there was no mandate in place to complete MEL activities—however, this approach was necessary due to the context in which this work was delivered. Satterlund et al. [44] also note that once relationships with staff are more established, support can be better tailored to local project needs, due to a better understanding of the local context. The building of collaborative relationships can also have snowball effect on fostering engagement with other programs/staff [44], and this has been seen in Brophy's broader capacity building program (not reported here). The strengthening of such relationships paired with an understanding of the context in which the work is being performed can facilitate the ability to adapt and change the required support based on a greater understanding of a project's objectives and goals.

The need to adjust and modify an intervention based on ongoing learning about contextual factors accords with the Dynamic Sustainability Model [48]. This model suggests that the implementation of interventions should be guided by continued learning and problem solving; the ongoing adaption of interventions to concentrate on their fit between the intervention, practice setting (context), and ecological system context; and expectations for ongoing improvement [48]. The model argues that change is constant across each of these levels, and accordingly, the successful sustainment of an intervention needs to consider measurement, negotiation, and the reciprocal fit of an intervention across these settings. This model also rejects the assumption that interventions can be optimised before they are implemented because of the importance of local context to optimal settings. Indeed, this has been seen in the current evaluation, whereby although adaptions to the number of outcome measures and data collection methods used (i.e., intervention context) have been made, these changes have occurred as a result of ongoing learning about the local context and the needs of those who are engaged in the work (i.e., practice setting and ecological system contexts). This has resulted in an improvement in the integration of the MEL pilots into the ongoing service delivery process. Moreover, these adaptations have been facilitated by the willingness and flexibility of MEL pilot teams who were open to making adjustments to implementation even when this resulted in additional time being spent on this work.

Organisational culture and climate structure sit within the practice setting context of the Dynamic Sustainability Model and are important components of intervention sustainability [48]. Organisational structure and attitude can facilitate evaluation capacity building [22]. This includes Executive buy-in, evaluation being an organisational priority, increased communication about evaluation, and the demonstration of strong evaluation leadership through the embedding of such work in processes, policy, and procedures. The current formative evaluation highlights that a team subculture is not adequate to fully support an evaluation culture and that the early stages of this work may have not allowed enough time to diffuse evaluation being an organisational priority to the whole staff cohort, potentially limiting the perceived value of this work and affecting the institutionalisation of organisational learning.

Importantly, evaluation capacity building can take several years to come to fruition [49]. This conflates with the recognition that outcomes for human services can take years to manifest [6], and as such, it may be difficult for CSO staff to

see the value of such activities in the initial years. In addition, the social work industry experiences high staff turnover, increased workloads, and inadequate staffing levels to meet demands [50]. This may also be the case for rural CSOs who—alongside rural healthcare organisations—may have difficulty attracting and retaining staff, making it challenging to implement initiatives that necessitate longer time commitments, such as evaluation capacity building. Indeed, the barriers seen in the current study, including staff having limited capacity to dedicate to MEL, having additional work responsibilities, and staffing changes may be a direct result of these wider constraints to both the sector and the rural context.

Nonetheless, having an organisational expectation and mechanism which utilises evaluation outcomes to inform practice can foster buy-in from staff [22], and the need for a cohesive organisational mission to galvanize employees has been identified in the research on quality improvement in healthcare organisations [51] and may be applied to the CSO setting. The ability to influence organisational structure with research and evaluation activity may be easier in smaller rural organisations, where there may be more scope to demonstrate immediate impact [52]. This has been demonstrated in Colac Area Health—a medium rural town [MM 4] [34], located just over 100 km from Warrnambool—where the organisation's first internally-driven research project led to increased research capacity, a sense of achievement amongst staff, and research being seen as a usual part of business through the establishment of a research unit [53].

The difficulties we observed with implementing this change initiative is common across sectors, and such challenges tend to persist regardless of improvements in organisational measures, enhancements to culture, advances in decision making, or the inclusion of change management strategies [54]. Within both the Australian and global nonprofit sector, a myriad of challenges exist which affect strategic and operational capacity, leading to difficulties with an organisation's ability to expand [55]. Such issues include challenges with revenue generation, performance management, leadership and management ideologies, and governance [55]. Further, the demanding environment in which the nonprofit sector exists endangers program funding and places increased demands on service delivery [56], which may further limit capacity for specific change initiatives. These challenges tend to represent practice setting and ecological system contexts of this work [48]. If resource-constrained CSOs are expected to do more (increased reporting) without increased funding, it seems reasonable to anticipate increased tension for change around capacity building if such innovation draws on scarce resources. This has the potential to hinder capacity building. Development of a shared understanding of evaluation as part of the organisational culture may help to mitigate such tension in CSOs and should be the focus of further work.

Of the limited literature on change initiatives in the Australian nonprofit sector, Rosenbaum et al. identified that reflection is a pre-condition to success, in that it is critical to permit sufficient time for all members involved in a change initiative to absorb and understand the initiative [55]. This reflection should occur at all stages of implementation, and adequate planning time should be included prior to implementation to allow this to occur. Further, Rosenbaum et al. identified that confidence and trust in the change initiative, focussing on the individual as well as the organisation, and timing attributes for planning the change (i.e., focus, design, delivery, frequency, and content of communication) influence the outcomes of change initiatives within nonprofit organisations [55]. As such, additional planning time to demonstrate the value of the MEL pilots to all relevant staff (i.e., both those involved in the MEL pilot teams, and those involved in data collection who were not part of the initial MEL pilot team) may have increased confidence and investment in the change initiative. This is echoed in Liao and colleagues' position regarding non-managerial employees being central to the adoption and implementation of change initiatives—particularly concerning employee value-alignment to organisational strategy [57]. Further, the pace and sequence of change initiatives must be considered to prevent fatigue and cynicism [57]. Focussing on fewer outcomes and data collection methods going forward may thus allow greater investment and resources to be concentrated on select outcomes, facilitating greater buy-in from staff.

Moreover, although the Dynamic Sustainability model argues that the optimal fit of interventions should be consistently tracked through the use of valid, reliable, and relevant measures [48], the complexity of human services restricts the ability to use standardised measures and aggregate evidence at the organisational level. This was also reflected in our difficulty in locating a suitable, standardised tool to measure evaluation capacity building amongst staff in CSOs. The use of novel

frameworks (i.e., the Community Services Outcomes Tree 30) allow a diverse range of outcomes that are common to CSOs to be defined, measured, and aggregated—which have been challenges to date [6]—and such frameworks may be suitable for other CSOs to consider using going forward.

Despite the challenges of this work a number of facilitators buoyed implementation. This included the physical presence of the Research and Evaluation Lead, which allowed informal discussions and on-the-spot problem-solving with MEL pilot teams to occur. This was especially the case for the two pilot teams who were located within the same building as the Research and Evaluation Lead (while the other teams were located separately, including some staff members who worked in offices >100km away). Communication to support staff also facilitated implementation. It is noteworthy that requests for the Research and Evaluation Lead to meet with wider program teams, as well as the creation of additional bespoke materials to support data collection (i.e., 'how-to' documents, printable QR codes, and simplified theory of change documents for wider program staff) was provided to pilot teams who had closer physical proximity to the Research and Evaluation Lead—aligning with the embedded research literature detailing that physical presence, paired with trusted relationships developed through a sense of 'embeddedness' can support staff in participating in research and evaluation activities [25].

## Reflections

The first author (BEK) works as an embedded researcher at Brophy. While it is a privilege to be a team member and conduct research both with Brophy and on the implementation of MEL at Brophy, this dual role created a power imbalance. This power imbalance stemmed from the embedded researcher towards the participants due to the inherent influence that an embedded researcher has over the research or evaluation design and interpretation. This power imbalance likely affected participant engagement with the follow-up survey and interviews in the current research, particularly if participants did not feel comfortable to report on their experiences of the MEL pilot implementation. In addition, at times, it was a challenge to navigate the different cultures of academia and the CSO, whereby academia is acutely focussed on producing research output, whereas this is novel for CSOs. As such, this change of pace created tension for the embedded researcher. These experiences undoubtedly shaped the current research.

## Strengths and limitations

Strengths of this formative evaluation include efforts to develop, implement, and measure a process for piloting MEL in a medium-sized rural CSO with limited resources to dedicate to such work. There are limited other settings where it would be possible to consider such work. Further, bespoke approach to each team's involvement in MEL considered each team's desires and resources, however, this in turn, limited an ability to standardise measures and team protocols. Nonetheless, this reflects the contention that evaluation capacity building is an inexact science [44]. Lastly, the local partnership between Brophy and the Australian Government's Rural Health Multidisciplinary Training program (RHMT) (see Kavanagh et al for further information 35)permits the potential to scale this work across other University Departments of Rural Health.

Limitations of this work include the small scale and limited timeframe in which this work was conducted within. Further, an appropriate, standardised measure to determine evaluation capacity building amongst staff in CSOs could not be identified. Instead, a tailored pre- and post-survey was developed, however, the tool was limited in its conceptual design which may affect its reliability and validity. Nil responses to the follow-up survey and limited interview data precluded an accurate interpretation of the effect of the MEL pilots on staff awareness, beliefs, attitudes, knowledge, skills, and implementation, as well as the broader experiences of the MEL. These data gaps may affect the validity of our conclusions as we cannot accurately contextualise participants' first-hand experiences of being part of this evaluation capacity building program. Importantly, the results of this work are largely based on administrative data, as well as the first-author's experience in coordinating the MEL pilots. As such, the results may be specific to the context of the current study and may not

be generalisable to other settings. Interestingly, poor questionnaire response has been reported in other studies on evaluation capacity building [58,59]. Efforts to increase data collection will be made in future. This may include the embedding of surveys during workshops or meetings and brief intercept interviews. Finally, although an important factor for initiating MEL, members of the four pilot teams were selected based on their role and interest in MEL, and as such, this may have introduced a positive bias towards conducting this work. This approach is in line with Roger's diffusion of innovations theory, which suggests that capable and committed 'innovators' and 'early adopters' undertake initial innovation cycles [60]. However, further work needs to be undertaken to better support organisation-wide uptake of MEL and learning from these initial cycles.

## Conclusions

In conclusion, this formative evaluation revealed that although the initial interest and buy-in from staff to engage in MEL was high, several barriers led to a divergence in the planned protocols for such work. For all teams, this included a divergence from the planned outcome measures. As a result, three pilot team have since scaled back the number of outcome measures being used for data collection, and a greater focus on the inclusion of the wider program team members in MEL decision-making. Further, since the commencement of the MEL pilots, an organisational research and evaluation strategy has subsequently been developed to further support the focus and research and evaluation activities, and to facilitate research and evaluation capacity building. Enablers to this work included physical presence to facilitate informal discussions, being flexible and open to changes, and written and verbal communication to aid staff understanding in the process and requirements of the MEL pilots. Implications of this formative evaluation include the ability of this work to be used as a blueprint for other organisations to conduct MEL activities and build a program of evaluation capacity building within the community service sector, paying particular attention to the balance and interest of staff and the challenges of embedding MEL into routine practice.

## Supporting information

**S1 Table.  Dimension and explanation of RE-AIM framework [39].**
(DOCX)

**S1 File.  Baseline and follow-up questionnaires.**
(DOCX)

**S2 File.  Semi-structured interview guide.**
(DOCX)

**S3 File.  Post training survey results.**
(CSV)

## Acknowledgments

The authors would like to acknowledge Brophy's Executive Leadership team for their support of this work, as well as Brophy's wider staff cohort for contributing their time and effort to undertaking MEL activities.

## Author contributions

**Conceptualization:** Bianca E Kavanagh, Vincent L Versace, Carly Dennis, Kevin Mc Namara.

**Data curation:** Bianca E Kavanagh.

**Formal analysis:** Bianca E Kavanagh.

**Funding acquisition:** Vincent L Versace, Kevin Mc Namara.

**Investigation:** Bianca E Kavanagh.

**Methodology:** Bianca E Kavanagh.

**Project administration:** Bianca E Kavanagh, Carly Dennis.

**Resources:** Bianca E Kavanagh.

**Supervision:** Vincent L Versace, Carly Dennis, Kevin Mc Namara.

**Visualization:** Bianca E Kavanagh.

**Writing – original draft:** Bianca E Kavanagh.

**Writing – review & editing:** Bianca E Kavanagh, Vincent L Versace, Hannah Beks, Carly Dennis, Marley J Binder, Andrea L Hernan, Kevin Mc Namara.

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
