## [Decision Letter · Decision Letter 0]

PONE-D-24-58659Evaluation capacity building in a rural Victorian community service organisation: A formative evaluationPLOS ONE

Dear Dr. Kavanagh,

Thank you for submitting your manuscript to PLOS ONE. After careful consideration, we feel that it has merit but does not fully meet PLOS ONE’s publication criteria as it currently stands. Therefore, we invite you to submit a revised version of the manuscript that addresses the points raised during the review process.

Both reviewers have provided detailed comments about your manuscript, which we would appreciate you addressing in full. One thing I would like to highlight is Reviewer 1's comments about anonymity for the case study institution - is it necessary to name the institution? It's not clear in the ethical approval if this is necessary or not; however, some clarity on whether the institution is happy to be named would be beneficial. My assumption is by partaking in the study they were happy with this, but it would be good to clarify.

We look forward to receiving your revised manuscript.

Kind regards,

Daniel Parkes, PhD

Staff Editor

PLOS ONE

Journal Requirements:

Reviewers' comments:

Reviewer's Responses to Questions

**Comments to the Author**

1. Is the manuscript technically sound, and do the data support the conclusions?

Reviewer #1: Yes

Reviewer #2: Partly

2. Has the statistical analysis been performed appropriately and rigorously? 

Reviewer #1: I Don't Know

Reviewer #2: N/A

3. Have the authors made all data underlying the findings in their manuscript fully available?

Reviewer #1: Yes

Reviewer #2: Yes

4. Is the manuscript presented in an intelligible fashion and written in standard English?

Reviewer #1: Yes

Reviewer #2: Yes

5. Review Comments to the Author

Reviewer #1: The article is interesting,timely and well presented. Here are some aspects that could be clarified:

* The MEL concept (page 5) needs to be presented a little bit more as it is a concept that appears in both research, policy documents and ngo reports. Some say Measurement, not monitoring etc.

* The authors could show how this topic is relevant across geographical and professional boundaries as it might seem a bit 'limited' to the Australian context and to the local SCOs there.

* In line with this: the authors could show more familiarity with research in the field, e.g. when they show the relevance of an embedded researcher, they could also link to this to major discussions in the field regarding the need for research-minded practitioners. For example as Mike Austin, Berkeley and Bowen McBeath, Portland USA has published about.

* The authors could also refer a bit more to the extensive literature on barriers to organizational learning to avoid that the discussion and "problem" is only about experiences from this particular context. The article has the potential to be an important addition to this research on the international context, and should therefor not be limited only to the Australian and this particular rural setting. The topic is familiar across professions and globally. Also literature on evaluation research could be mentioned (evaluation as research field)

STUDY CONTEXT (page 6-7) - here some clarifications are needed:

But first of all: The place is not anonymized. Is that ok by the informants? The authors should reflect on this.

1. Are the authors both employed at and researching in their own working place and among their collegas? That must be reflected on if that is the case, as it may affect biases in data collection, mixed roles etc.

2. Who designed the program og evaluation capacity (page 6, line109)?

3. Explain the selection criteria: Why the 4 programs were selected (page 7, line118), what area do they cover? And why these four of all 38 programs).

4. What do you mean by 'the wider program- (page 7, line 119)

5. Members of the teams were selected 'based on their role and interest' (page 119-120). How can this affect the data and results? (Will they be more positive towards the research?)

6. Were clients involved at any stage of this research process (why/why not)?

7. The sentence on page 8 line 147-149 is nearly identical with the sentence on Material and Method on the same page, line 157-159, and should be deleted one of the places.

MATERIAL AND METHODS

1. Consent is described, however the ethical challenge on not anonymizing the place must be reflected on. Do the workers know, and is it ok? What are the challenges of not covering where this research is taking place?

2. The qualitative data - how are they being used in the Results section? How are they analysed? The section on Data analysis need to be strengthened. It is only 5 lines, and in the name of transparency in research, i would be helpful if the authors could explain in some more detail how they analysed the diverse data collected by ethnographic and interview methods. And to show clearer how it is used and presented in the Results and Discussion.

3. The information of the informants' varied background on page 13-14 (lines 234-238) I recommend that is moved to the Study context section where data and selection is mentioned.

RESULTS

Some parts could be considered moved to the Discussion/ Implications. For example the reflection on page 18 lines 302-309.

I miss some results from the qualitative data

DISCUSSION

The section sums up the study and facilitators/barriers well, and succeeds in showing why this under-researched topic is crucial for the enhancement for local services. They also bring interesting topics to the discussion regarding how to develop capacity building for this work. The important potential of a link officer, embedded researcher, developing research-minded practitioners etc is of high interest in this paper.

The dynamic sustainability model is introduced which contributes to the discussion. It is good that the authors finally links their discussion to a wider context (page 21).

The section is quite long, and some of its parts is about lessons learned, and consequently, implications for further practice. These parts could be structured and presented in a separate section following the Discussion (e.g. development of shared understanding, how to facilitate for more reflection as a pre-condition for success etc.)in

Limitations and strengths could also be presented in a separate section.

Thank you for the opportunity to read the interesting study and good luck in finalizing the article.

Reviewer #2: The manuscript presents valuable and original research regarding evaluation capacity building (ECB) within a rural, place-based community service organisation (CSO) in Victoria, Australia. Focusing on the early implementation of Monitoring, Evaluation, and Learning (MEL) pilots, the study addresses a documented gap in the literature, particularly for rural and resource-constrained settings. The use of the RE-AIM framework to structure this formative evaluation is appropriate and well justified. The paper is timely, aligning with current scholarly interest in strengthening ECB practice, as demonstrated by a recent special issue of New Directions for Evaluation (2024, Issue 183) dedicated entirely to exploring best practices and effective methodologies for evaluating ECB itself

While the topic is important and the initial work valuable, major revisions are required to improve methodological transparency, address the data gaps appropriately, and ensure conclusions are robustly supported by the available evidence. I offer the following suggestions for the authors’ consideration:

General and Conceptual Suggestions

- The study makes a strong case for the importance of MEL in CSOs, but the link between rurality and the specific challenges of MEL implementation could be strengthened. The Modified Monash (MM) classification is used to define rurality, yet the implications of this classification for workforce, infrastructure, and program delivery are not clearly connected to the evaluation design or the observed challenges.

- The manuscript notes that Brophy identified a need for MEL integration; however, it would be helpful to clarify how and by whom this need was identified—e.g., via leadership, internal audit, funder requirements, or staff demand.

- The pedagogical approach to MEL training could be described more clearly. The authors imply an experiential learning model (participants applied tools to their own programs), but this is not explicitly stated. Clarifying whether the approach was experiential, practice-based, or otherwise grounded in adult learning theory would help contextualize the training’s design and intent.

- The authors mention that five MEL modules were delivered between June and October 2023. Specifying the approximate time commitment (e.g., total hours, duration of sessions) and whether participants received materials such as workbooks, templates, or pre/post tasks would be useful.

- The “evaluation champions” strategy is briefly noted as an implementation enabler. It would be helpful to indicate whether this was a formal designation or an informally observed role, and what functions these champions were expected or observed to play in supporting MEL adoption.

Methods & Results

- The supplementary materials, including the RE-AIM operationalization (Supplementary Material 1), questionnaire (Supplementary Material 2), and interview guide (Supplementary Material 3), were reviewed and provide useful context for the study methods (though the questionnaire warrants revision as noted below).

- The first author was embedded in the organization and facilitated the intervention. While this position likely offered advantages, the discussion does not adequately reflect on potential biases or power dynamics that may have influenced participant responses or engagement. A brief reflexive comment on this dual role would add methodological integrity.

- While the use of the RE-AIM framework is a strength, some domains are addressed more robustly than others. For instance, “Reach” is primarily described in terms of team selection, with limited analysis of participant engagement or reasons for non-participation. Similarly, “Implementation” focuses mainly on fidelity to the protocol (e.g., number of outcomes developed), but does not address the cost or resource requirements of the MEL intervention—an element sometimes included within the RE-AIM framework (and noted as potentially relevant by the authors in Supplementary Material 1). The authors should clarify which RE-AIM domains were prioritized or fully evaluated at this early implementation stage.

- The manuscript would benefit from a clearer explanation of how data source triangulation was undertaken. While the study is described as using a mixed methods approach—including self-report questionnaires, administrative data, and (planned) qualitative interviews—it remains unclear how these data sources were compared, integrated, or analyzed in relation to one another. For example, were questionnaire findings cross-validated against administrative records or implementation logs? Were efforts made to identify convergence or divergence across data types? The strength of a mixed methods design often lies in triangulating multiple perspectives or forms of evidence. In its current form, the manuscript does not fully demonstrate how this integration occurred, weakening the mixed methods claim. Brief clarification would enhance methodological transparency and credibility.

- A self-report questionnaire (Supplementary Material 2), adapted from a tool on evidence-informed practice, was used to assess MEL-related constructs (e.g., awareness, beliefs, attitudes, knowledge, skills, implementation). However, the tool appears to have several weaknesses. Firstly, several items seem to measure overlapping constructs—e.g., “beliefs” (that MEL improves outcomes) and “attitudes” (willingness to support MEL)—which may limit conceptual distinction. Secondly, the tool mixes response formats (Likert scales, dichotomous items), which can undermine consistency and comparability. Thirdly, based on the items provided, it appears each dimension (Awareness, Beliefs, Attitudes, Knowledge, Skills) is assessed using only a single item. Relying on single-item measures can significantly limit reliability and validity compared to multi-item scales. The authors should clarify how the original tool was adapted for this study and describe the adaptation process, acknowledging the limitations of the measurement approach used. Future iterations should consider a more rigorously developed instrument, potentially using multi-item scales aligned with established ECB frameworks (e.g., Preskill & Boyle, Nielsen).

- The data collection using administrative records is appropriate, but the authors could better explain how these data were analyzed—e.g., what criteria determined protocol adherence, and how deviations were classified.

- The description of short evaluation cycles (SECs) would benefit from more detail. How were lessons from each SEC integrated into MEL refinements? Including an example of how adaptations occurred in real time would reinforce the developmental nature of the work.

- The participant sample size for the questionnaires is very limited (n=6 post-training, n=0 follow-up). Consider discussing whether additional data collection efforts (e.g., reminders, in-person data capture, brief intercept interviews) were attempted or might be feasible in the future.

- Several implementation barriers are reported (e.g., limited staff capacity, organizational change, unclear value of MEL), but it is often unclear which data source supports each conclusion. Were these insights drawn from survey responses, meeting logs, informal reports, or researcher observation? For example, the barrier “waxing and waning engagement” would be more compelling if grounded in specific data (e.g., frequency of missed meetings, analysis of communication logs).

Discussion

- The manuscript includes a theory of change model (Figure 1), but the discussion does not return to it sufficiently. There is limited reflection on whether initial assumptions about mechanisms of change were confirmed, challenged, or refined based on the evaluation findings. This is a missed opportunity, especially in a formative evaluation context.

- While many barriers are discussed, enabling factors (e.g., physical presence of the evaluation lead, supportive leadership, flexible design) are mentioned relatively briefly. A more balanced discussion synthesizing what worked could be useful for practitioners reading this paper for guidance.

- Although the authors acknowledge the low response rate and lack of interview data in the Results section, these limitations are not sufficiently explored in the Discussion. The manuscript requires a more critical reflection on how these data gaps affect the credibility, depth, and validity of the conclusions. For instance, the absence of follow-up survey responses prevents robust assessment of changes in MEL-related knowledge, attitudes, or practices over time. Similarly, the exclusion of qualitative interview data—originally planned to contextualize implementation challenges—leaves an important gap in understanding participants’ perspectives. The authors should more clearly articulate how the small sample and missing data limit the robustness and generalizability of the findings. They could also suggest realistic data collection strategies for future iterations (e.g., embedded follow-up questions during workshops, brief written reflections, group interviews, or analysis of participant-generated outputs such as MEL plans) to help mitigate such issues. Strengthening this aspect of the discussion would improve transparency and help other practitioners or researchers facing similar constraints.

Overall, this study addresses an important topic in a relevant context. If the authors can address the methodological and reporting issues outlined above, particularly regarding the data limitations and transparency, the manuscript could make a valuable contribution to the literature on evaluation capacity building in community service organisations.

6. PLOS authors have the option to publish the peer review history of their article (what does this mean? ). If published, this will include your full peer review and any attached files.

**Do you want your identity to be public for this peer review?** For information about this choice, including consent withdrawal, please see our Privacy Policy .

Reviewer #1: No

Reviewer #2: **Yes: ** David Buetti

---

## [Author Response · Author response to Decision Letter 1]

6 Jun 2025

Wednesday, 4 June 2025

Professor Daniel Parkes

Staff Editor, PLOS ONE

RE: Revision of manuscript PONE-D-24-58659 “Evaluation capacity building in a rural Victorian community service organisation: A formative evaluation”.

We thank the reviewers for their feedback and for the opportunity to revise our manuscript. We have now addressed the queries raised by the reviewers. In this document, the reviewer’s queries are presented in boldface and our responses are in regular text, with changes to the manuscript text indicated in italics and quotation marks. In the revised manuscript submission, changes from the original manuscript are displayed using yellow highlighted text.

We also sought confirmation from the case study organisation and can confirm that the organisation is happy to be named in the manuscript.

Kind regards

Dr Bianca E Kavanagh

Deakin Rural Health

School of Medicine – Faculty of Health, Deakin University

Reviewer’s comments to author

Reviewer 1

The article is interesting, timely and well presented. Here are some aspects that could be clarified:

The MEL concept (page 5) needs to be presented a little bit more as it is a concept that appears in both research, policy documents and ngo reports. Some say Measurement, not monitoring etc.

The MEL concept, including clarification of the terminology of ‘monitoring’ vs ‘measurement’ has been expanded.

“MEL is utilised across many organisational contexts, including in policy documents, non-governmental reports, and research. In a diverse range of settings including CSOs, agriculture and international aid, MEL supports increased accountability, adaptability, change optimisation and learning where it is desirable to understand the impact of innovation and knowledge translation activities designed to support strategic organisational and system changes (14–16). Some MEL frameworks utilised by organisations are made publicly available to support transparency (e.g., 11,12). Oftentimes, the term ‘monitoring’ is used interchangeably with ‘measurement’, however, monitoring may be perceived as being more active in its approach to using and learning from data insights, and thus is the approach discussed in this paper.”

The authors could show how this topic is relevant across geographical and professional boundaries as it might seem a bit 'limited' to the Australian context and to the local SCOs there.

We feel that with the addition of the suggested literature on practice research and embedded research has broadened the relevance of this topic across geographical and professional boundaries (see response to the below comment). We have also included a sentence outlining that the CSO sector may utilise lessons learned from social work.

“…and this sector may benefit from lessons learned in the broader social work sector.”

In the introduction we have also identified that MEL is used in a variety of policy translation and change initiatives across different NGO sectors.

In line with this: the authors could show more familiarity with research in the field, e.g. when they show the relevance of an embedded researcher, they could also link to this to major discussions in the field regarding the need for research-minded practitioners. For example as Mike Austin, Berkeley and Bowen McBeath, Portland USA has published about.

Thank you for the suggestion to include research on practice research and embedded research. We have now included information on these topics in text.

“An emerging area which may be beneficial to MEL and evaluation capacity building is the implementation of practice research or embedded research. Practice research is entrenched in social services (16), whereby social work researchers and practitioners work together to identify, understand, and improve service delivery and organisational structures and processes. McBeath et al (17) recommend that collaborative, trust-based relationships should be developed and sustained to achieve practice research. Further, sustained self-reflection including deliberation of the tension between the type of research generated (e.g., empirical peer-reviewed research vs grey literature, agency-based statistics) as well as the expectations of organisational staff, service users, and other stakeholders should occur to maintain varied relational commitments (17). Embedded research also aims to build the capacity of an organisation to conduct research via the embedding of a researcher into a host organisation. The embedded researcher is generally required to developed relationships with staff at the host organisation and be seen as a member of the team and to generate knowledge that meets the needs of the host organisation(18). Embedded research, however, has traditionally been conducted within healthcare settings, and is an emerging area of research in other settings, including within CSOs.”

“To support this, CSOs must have research-minded practitioners who are interested in and utilise evidence to inform practice. Research practitioners or embedded researchers may support practitioners to become research-minded, through their assistance and leadership in developing processes and building capacity to collaboratively evaluate processes and create practice-based solutions (21,22).”

* The authors could also refer a bit more to the extensive literature on barriers to organizational learning to avoid that the discussion and "problem" is only about experiences from this particular context. The article has the potential to be an important addition to this research on the international context, and should therefor not be limited only to the Australian and this particular rural setting. The topic is familiar across professions and globally. Also literature on evaluation research could be mentioned (evaluation as research field)

We agree that information on organisational learning helps to broaden the potential context of this paper. Information on organisational learning, as well as evaluation research, has now been included in text.

“It has also seen the need to create and share organisational knowledge, however literature on organisational learning and knowledge is an underrepresented area in the literature for not-for-profit organisations (9). Organisational learning refers to the collective learning process that is regulated in an organisation to improve organisational performance and goals, affecting an organisation’s processes, structures, routines, and future learnings (10). Crossan et al’s model organisational learning suggests that it is dynamic in nature; integrates individual, group, and organisational levels of analysis; incorporates exploration and exploitation tensions; and is open to various types of changes grounded in experience (11). The model suggests that organisational learning occurs via bidirectional intuiting, interpreting, innovation, and influence processes that operate across the individual, group, and organisational levels. Barriers to organisational learning can therefore be targeted towards these processes (10).”

“…and may also promote organisational learning.”

“…and affecting the institutionalisation of organisational learning.”

“Evaluation as a field of research is receiving increased recognition for its primary purpose of collating information with the ultimate use being in decision making (15).”

STUDY CONTEXT (page 6-7) - here some clarifications are needed:

But first of all: The place is not anonymized. Is that ok by the informants? The authors should reflect on this.

We can confirm that the organisation is happy to be named in the manuscript as it represents an important starting point in their journey to conduct and contribute to research and evaluation literature.

1. Are the authors both employed at and researching in their own working place and among their colleagues? That must be reflected on if that is the case, as it may affect biases in data collection, mixed roles etc.

Thank you for this suggestion. This reflection has now been added in text.

“The first author (BEK) works as an embedded researcher at Brophy. While it is a privilege to be a team member and conduct research both with Brophy and on the implementation of MEL at Brophy, this dual role created a power imbalance. This power imbalance stemmed from the embedded researcher towards the participants due to the inherent influence that an embedded researcher has over the research or evaluation design and interpretation. This power imbalance likely affected participant engagement with the follow-up survey and interviews in the current research, particularly if participants did not feel comfortable to report on their experiences of the MEL pilot implementation. In addition, at times, it was a challenge to navigate the different cultures of academia and the CSO, whereby academia is acutely focussed on producing research output, whereas this is novel for CSOs. As such, this change of pace created tension for the embedded researcher. These experiences undoubtedly shaped the current research.”

2. Who designed the program of evaluation capacity (page 6, line109)?

The program was designed initially by Brophy’s Executive Leadership Team, Quality and Strategic Projects Manager (CD), and Research and Evaluation Lead. Input was also received from Deakin Rural Health, as part of a partnership with the embedded research position.

“…by members of Brophy’s Executive Leadership Team, Quality and Strategic Projects Manager (CD), Research and Evaluation Lead. Additionally, members of Deakin Rural Health (a University Department of Rural Health) also had input into the monitoring and evaluation of the capacity building program.”

3. Explain the selection criteria: Why the 4 programs were selected (page 7, line118), what area do they cover? And why these four of all 38 programs).

The selection criteria were based on programs that had a need to capture outcomes for funding reports, were newly developed and delivered, or had a need to understand why and how outcomes were being achieved for clients. The Executive Leadership Team recognised the four selected programs as having these requirements, as well as having staff that had an interest in MEL. Information concerning the selection criteria and included programs have now been provided in text.

“(i.e., programs that were either newly delivered, had a need to better understand the outcomes their clients were achieving, or had funding requirements that necessitated data to be captured and reported). These programs included services for individual and family support, alcohol and other drugs, foster care, and youth homelessness and employment.”

4. What do you mean by 'the wider program- (page 7, line 119)

The wider program refers to select staff who are members of a broader team of a program of work (i.e., three members of staff who are taking part in the evaluation capacity building program who are part of a broader team of people working in a specific program of work). The wording has been clarified in text.

“(i.e., staff representatives who are taking part in the MEL pilots who belong to a wider team of staff who are employed within a specific program of work)”

5. Members of the teams were selected 'based on their role and interest' (page 119-120). How can this affect the data and results? (Will they be more positive towards the research?)

Members were selected based on their role, including their level of employment in the organisation (i.e., Executive Manager, Manger, Team member) and if they had shown interest in understanding client outcomes. Members were selected at the discretion of the Executive Leadership Team. It is likely that these members would be more positive towards research and evaluation due to their interest, however, this was an important factor to get the work of the ground. This approach is also in line with Roger’s Diffusion of Innovation Theory which suggests that people who are open to new ideas and taking risks will adopt first before such ideas are more widely diffused. We acknowledge that further work needs to be done to better support organisation-wide uptake and learning from these initial cycles.

“Although an important factor for initiating MEL, members of the four pilot teams were selected based on their role and interest in MEL, and as such, this may have introduced a positive bias towards conducting this work. This approach is in line with Roger’s diffusion of innovations theory, which suggests that capable and committed ‘innovators’ and ‘early adopters’ undertake initial innovation cycles (60). However, further work needs to be undertaken to better support organisation-wide uptake of MEL and learning from these initial cycles.”

6. Were clients involved at any stage of this research process (why/why not)?

Due to ethics limitations, clients were not involved in the research process. However, clients will be involved in future MEL work within the organisation and ethics approval is currently being sought for this work.

7. The sentence on page 8 line 147-149 is nearly identical with the sentence on Material and Method on the same page, line 157-159, and should be deleted one of the places.

Thank you for recognising this mistake. The sentence has been removed from the materials and methods section.

MATERIAL AND METHODS

1. Consent is described, however the ethical challenge on not anonymizing the place must be reflected on. Do the workers know, and is it ok? What are the challenges of not covering where this research is taking place?

The MEL pilot team members are aware that research on the evaluation capacity building program was taking place. All staff involved in the MEL pilot teams were sent emails describing the research and the potential to be involved in the questionnaires and interviews. Emails included the PICF which detailed the administrative data being collected. Staff were reminded in-person (using ethics-approved scripts) at relevant MEL meetings.

The challenges of disclosing where this research was conducted include that it may impact organisation privacy and has the potential to tarnish (or increase, in some cases) organisational reputation. However, the organisation is motivated to conduct, participate in, and disseminate research.

2. The qualitative data - how are they being used in the Results section? How are they analysed? The section on Data analysis need to be strengthened. It is only 5 lines, and in the name of transparency in research, it would be helpful if the authors could explain in some more detail how they analysed the diverse data collected by ethnographic and interview methods. And to show clearer how it is used and presented in the Results and Discussion.

Due to the lack of interviews conducted, nil qualitative interview data was included in the analysis. The analysis included quantitative data from the post training/refinement survey and administrative records. Minutes from meetings were analysed to identify common high-level themes relating to the implementation of the pilots.

“…and nil qualitative interview data was included in the analysis”

“Microsoft Excel, Forms, and List documents were assessed against each team’s protocol and measurement framework as having been administered according to planned timeline, outcome measures, and reporting requirements, and these items were discussed at MEL pilot meetings. High level themes that related to the implementation of the MEL pilots were extracted from meeting minutes. Adherence was measured as outcome measures being administered according to the protocol and measurement framework (yes/no). Deviations occurred when an outcome measure had not been administered at all. The number of meetings each pilot team had within the SECs was cross-tabulated and reported.”

3. The information of the informants' varied background on page 13-14 (lines 234-238) I recommend that is moved to the Study context section where data and selection is mentioned.

Thank you for this suggestion, however, the information on participants’ backgrounds in the Reach section includes only those who completed the post-training/refinement survey (n=6 of 12), and as such only their roles and training backgrounds are included—we felt it was important to describe these details within this sec

---

## [Decision Letter · Decision Letter 1]

Evaluation capacity building in a rural Victorian community service organisation: A formative evaluation

PONE-D-24-58659R1

Dear Dr. Kavanagh,

We’re pleased to inform you that your manuscript has been judged scientifically suitable for publication and will be formally accepted for publication once it meets all outstanding technical requirements.

Kind regards,

Haitao Shi

Academic Editor

PLOS ONE

Additional Editor Comments (optional):

Reviewers' comments:

Reviewer's Responses to Questions

**Comments to the Author**

1. If the authors have adequately addressed your comments raised in a previous round of review and you feel that this manuscript is now acceptable for publication, you may indicate that here to bypass the “Comments to the Author” section, enter your conflict of interest statement in the “Confidential to Editor” section, and submit your "Accept" recommendation.

Reviewer #1: All comments have been addressed

Reviewer #2: All comments have been addressed

2. Is the manuscript technically sound, and do the data support the conclusions?

Reviewer #1: Yes

Reviewer #2: Yes

3. Has the statistical analysis been performed appropriately and rigorously? 

Reviewer #1: Yes

Reviewer #2: Yes

4. Have the authors made all data underlying the findings in their manuscript fully available?

Reviewer #1: Yes

Reviewer #2: Yes

5. Is the manuscript presented in an intelligible fashion and written in standard English?

Reviewer #1: Yes

Reviewer #2: Yes

6. Review Comments to the Author

Reviewer #1: (No Response)

Reviewer #2: I believe all comments have been addressed by the authors, providing clarifications on ECB intervention methods, data collection, and other relevant details. This paper makes a significant contribution to existing ECB literature.

7. PLOS authors have the option to publish the peer review history of their article (what does this mean? ). If published, this will include your full peer review and any attached files.

**Do you want your identity to be public for this peer review?** For information about this choice, including consent withdrawal, please see our Privacy Policy .

Reviewer #1: No

Reviewer #2: **Yes: ** David Buetti

---

## [Editor Report · Acceptance letter]

PONE-D-24-58659R1

PLOS ONE

Dear Dr. Kavanagh,

I'm pleased to inform you that your manuscript has been deemed suitable for publication in PLOS ONE. Congratulations! Your manuscript is now being handed over to our production team.

Kind regards,

on behalf of

Dr. Haitao Shi

Academic Editor

PLOS ONE